# Molecular recognition of the native HIV-1 MPER revealed by STED microscopy of single virions

Pablo Carravilla [1], Jakub Chojnacki [2], Edurne Rujas[1], Sara Insausti[1], Eneko Largo[1], Dominic Waithe [2], Beatriz Apellaniz[1], Taylor Sicard[3,4], Jean-Philippe Julien [3,4,5], Christian Eggeling[2,6,7] & José L. Nieva[1]

Antibodies against the Membrane-Proximal External Region (MPER) of the Env gp41 subunit neutralize HIV-1 with exceptional breadth and potency. Due to the lack of knowledge on the MPER native structure and accessibility, different and exclusive models have been proposed for the molecular mechanism of MPER recognition by broadly neutralizing antibodies. Here, accessibility of antibodies to the native Env MPER on single virions has been addressed through STED microscopy. STED imaging of fluorescently labeled Fabs reveals a common pattern of native Env recognition for HIV-1 antibodies targeting MPER or the surface subunit gp120. In the case of anti-MPER antibodies, the process evolves with extra contribution of interactions with the viral lipid membrane to binding specificity. Our data provide biophysical insights into the recognition of the potent and broadly neutralizing MPER epitope on HIV virions, and as such is of importance for the design of therapeutic interventions.

[1] Biofisika Institute (CSIC, UPV/EHU) and Department of Biochemistry and Molecular Biology, University of the Basque Country (UPV/EHU), P.O. Box 644, 48080 Bilbao, Spain. [2] MRC Human Immunology Unit, Weatherall Institute of Molecular Medicine, University of Oxford, Oxford OX3 9DS, UK. [3] Program in Molecular Medicine, The Hospital for Sick Children Research Institute, Toronto, ON M5G 0A4, Canada. [4] Department of Biochemistry, University of Toronto, Toronto, ON M5S 1A8, Canada. [5] Department of Immunology, University of Toronto, Toronto, ON M5S 1A8, Canada. [6] Institute of Applied Optics Friedrich-Schiller-University Jena, Max-Wien Platz 4, 07743 Jena, Germany. [7] Leibniz Institute of Photonic Technology e.V., Albert-Einstein-Straße 9, 07745 Jena, Germany. Correspondence and requests for materials should be addressed to C.E. (email: christian.eggeling@rdm.ox.ac.uk) or to J.L.N. (email: joseluis.nieva@ehu.es)

The envelope glycoprotein (Env) of the human immuno-deficiency virus type-1 (HIV-1) embodies a common class I viral fusion machinery, but also configures diverse antigenic surfaces across the different viral clades, strains, and isolates[1,2]. Quaternary structure adjustments, heavy glycosylation, and intrinsic genetic variability of Env are thought to allow viral escape from neutralization by Env-specific antibodies[3,4]. Despite the effectiveness of these escape mechanisms, discovery in the 1990s of a handful of broadly neutralizing antibodies (bnAbs) proved the potential of humoral immunity to protect from HIV-1 infection[3,4]. Recent advances in the isolation of bnAbs with different specificities, together with their systematic structure–function analyses, further support the existence of a number of sites of vulnerability on the native surface subunit gp120, or on the interface between this and the transmembrane subunit gp41[5–8].

Here, we focus on a distinct site of vulnerability existing on the transmembrane Env subunit gp41: the membrane proximal external region (MPER)[5,7]. Two reasons explain the special interest on solving the mechanisms underlying the molecular recognition of this Env site, namely, the exceptional degree of conservation of the MPER epitope sequence, and the fact that its engagement with the bnAbs 10E8 and 4E10 results in one of the broadest HIV neutralization levels described so far (98% of viruses blocked in customary infectivity tests)[5,9].

The 10E8/4E10 epitope arranges onto a lateral face of the continuous helix connecting the Env ectodomain with the transmembrane domain (TMD) at the point where this structural element emerges from the lipid bilayer[9–11] (Fig. 1). However, a mechanistic understanding of MPER recognition by bnAbs has been hampered by the limited information available on its native antigenic structure: a molecular surface that lies in contact with the viral membrane at the base of the Env complex (Fig. 1). This structural complexity has proven challenging to reproduce by model systems amenable to biophysical and biochemical characterization, and precludes crystallization of MPER-containing Env specimens[8,12–14]. Even for the case of a solubilized Env-detergent complex used in single-particle cryo-electron microscopy (Cryo-EM) studies[15], it is unclear whether it would correctly recapitulate native Env MPER conformation in the viral lipid. Furthermore, in these studies, the Fab 10E8 was used to

enable purification, which does not provide an opportunity to understand the native MPER epitope unliganded, and the mechanism of recognition; it only provides a still view post-binding albeit with sub-nanometer details.

Some authors postulate that, due to steric occlusion, the MPER helix remains hidden within the native pre-fusion Env complexes on virions, becoming accessible for binding in the fusion-activated Env intermediates[16–20]. This assumption implies that, in contrast to epitopes on the solvent-accessible subunit gp120, the neutralization competent structure of MPER would exist transiently, and its accessibility be limited to the population of virions primed for fusion, greatly limiting antibody efficacy. According to this model, this limitation would be surpassed by a pre-attachment step to the viral membrane through antibody–lipid interactions. This pre-insertion step would increase local antibody concentration around Env, waiting for MPER to be exposed[5,16–19,21].

More recently, based on structural information derived from Cryo-EM reconstructions of MPER-TMD-containing Env-Fab complexes[15], and the reported demonstration that individual Env molecules are conformationally dynamic[22,23], a model for the 10E8-bound form of the Env trimer has been proposed. According to that model, gp41 MPER helices are accessible in one of the possible pre-fusion conformations of the native Env, while the overall organization of the surface subunit gp120 remains unchanged. 10E8 binding would keep these helices in an uplifted position[15]. This model puts forward the possibility that MPER is transiently, but intrinsically, accessible in the native Env complex (Fig. 1). However, as mentioned before, the fact that Cryo-EM experiments were done in detergent-solubilized Env proteins raised the possibility that the mixed phospholipid–detergent micelles did not properly mimic the accessibility of the antibody to native MPER.

Here, we have assessed the accessibility of MPER on intact HIV-1 particles, i.e., in the context of membrane-anchored native Env complexes in the extraordinary HIV-1 lipid environment[24], using super-resolution-stimulated emission depletion (STED) microscopy. In the past, this optical microscopy technique has made possible sub-diffraction (~40 nm) observations of Env molecules on individual HIV particles (~120 nm in diameter) using anti-gp120 antibodies as reporters[25–27], and therefore constitutes a suitable tool to qualitatively and quantitatively measure anti-Env antibody binding to viruses. The examination of anti-MPER bnAb binding to intact virions through STED microscopy ruled out the existence of a membrane-attached, free antibody population, while providing evidence to support a correlation between the degree of binding to native Env and neutralization. Thus, Env-mediated association with virions appears to be applicable to all anti-HIV bnAbs tested, independently of potency, polyreactivity, or the targeted Env subunit. Furthermore, our data suggest that the strength of interaction with Env and neutralization potency evolve with extra contribution of antibody–membrane interactions in the case of the anti-MPER bnAbs 10E8 and 4E10.

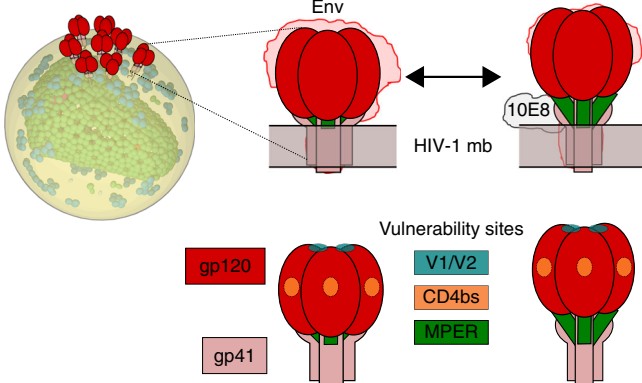

**Fig. 1** Model for MPER accessibility within native Env complexes based on Cryo-EM reconstructions[15]. Top: Contours derived from detergent-solubilized Env trimers, without (left) or with 10E8 bound (right), have been docked into a viral lipid bilayer. A putative transition between these two states would result in exposure of the MPER helix in native Env (green). Bottom: cartoons to designate in the previous models the different trimer components, and positions of the neutralizing epitopes for bnAbs used in this study PGT145 (V1/V2), VRC01 and b12 (CD4bs); and 10E8/4E10 (MPER)

## Results

**Antibody accessibility to native Env measured by STED**. To analyze MPER accessibility through STED microscopy in the native Env complex, 10E8 and 4E10 Fabs were conjugated with the fluorescent probe Abberior STAR RED (also known as KK114). To that end, we followed a site-directed conjugation strategy, which resulted in roughly equimolar Fab-to-K114 ratios, and labeling at a position irrelevant for Fab-epitope binding (see Methods). The produced Fab10E8-KK114 and Fab4E10-KK114 conjugates retained the neutralization activity of the unmodified

Fabs (Supplementary Table 1), confirming no-interference of the labeling process with functional binding to Env.

Figure 2a displays confocal images of eGFP-labeled NL4-3 viral particles incubated with KK114-labeled Fab 10E8, the latter visualized in the confocal, or in the super-resolved STED microscopy mode (red signals in top and bottom panels, respectively). The STED image reveals the KK114 signal confined within restricted areas on the virion surface, following the pattern described for mature HIV particles incubated with the anti-gp120 antibody 2G12[25,26]. Thus, binding of 2G12 to gp120, revealed by a secondary fluorescent antibody, and binding of directly labeled 10E8 or 4E10 to MPER, reflected similarly the clustering of native Env proteins on the virions (Fig. 2b, c).

Next, we assessed the capacity of anti-MPER bnAbs to directly bind to the viral membrane, i.e. within virion areas devoid of Env clusters. We compared the staining patterns induced by 2G12, 10E8, and 4E10 antibodies with those of phosphatidylserine (PS)-binding Annexin V-ATTO 647N using STED microscopy (Fig. 2b). Metabolically inert virions lose the aminophospholipid asymmetric distribution existing at the plasma membrane of producing cells, resulting in exposure of PS at the viral membrane external leaflet[28,29]. Consistent with its binding to the lipid component of the viral membrane, the super-resolved signal of the PS-binding protein Annexin V-ATTO 647N distributed over the complete surface of the virions characterized by eGFP signal (Fig. 2b). In contrast to Annexin, staining by 2G12 or the anti-MPER antibodies 10E8/4E10 occupied <20% of the viral area.

Furthermore, similarly to 2G12[25], both anti-MPER antibodies 10E8 and 4E10 in association with individual eGFP-labeled particles distributed predominantly into one single foci (ca. 80% of the particles), and less frequently into two, or more than two foci (Fig. 2c). Reflecting the consistency of these estimations, higher concentrations of Fabs 10E8 or 4E10 yielded a similar number of Env clusters per virion (Supplementary Fig. 1a). The foci distribution pattern detected for the KK114 conjugates was also reproduced by a set of anti-gp120 antibodies, and by anti-MPER antibodies when revealed by a secondary fluorescent antibody (Fig. 2d). These observations suggest that anti-MPER and anti-gp120 antibodies engage with the same type of Env clusters on the viral particles and argue against a membrane pre-

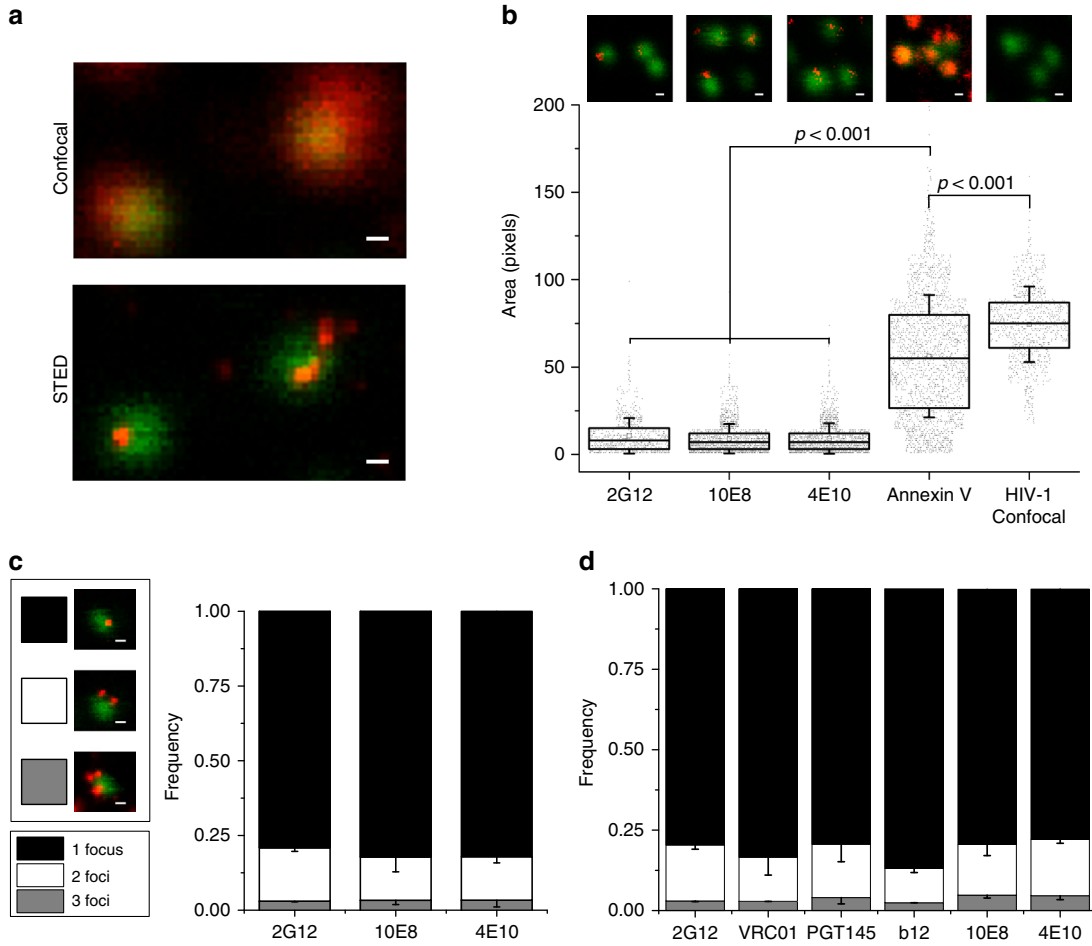

**Fig. 2** Accessibility of MPER Env site on intact HIV virions visualized by super-resolution STED microscopy. **a** Signal of 10E8-KK114 (red) bound to HIV-1 virions acquired in confocal (top) or STED modes (bottom). In both images virions are visualized in confocal mode (green, Vpr.eGFP). **b** STED micrographs of 2G12 (stained with secondary antibody), 10E8-KK114, 4E10-KK114, and Annexin V-ATTO 647 (STED, red) bound to HIV-1 virions (top) and corresponding pixel areas (bottom). Results are shown in a box-plot (center line, median; square, mean; box, interquartile range (IQR); whiskers, s.d.) of STED (2G12, $n = 756$; 10E8, $n = 1693$, 4E10, $n = 1575$; and Annexin V, $n = 2136$) and confocal (Vpr.eGFP, $n = 942$) signals measured in single virions from three independent preparations. Pixel size is 20 nm/pixel. The statistical significance was assessed by Kruskal–Wallis one-way analysis of variance. **c** Representative images of HIV virions incubated with 10E8-KK114 (left) and distribution analysis of Fab STED foci number (right) upon incubation with 2G12 (stained with secondary anti-human Fab-KK114) or directly labeled 10E8 and 4E10 Fab-KK114 conjugates. **d** Distribution of Fab STED foci detected on virions. Bound Fabs were revealed in this case by secondary staining (anti-human Fab-KK114). Error bars are s.d. of at least three independent experiments. Scale bars are 100 nm

attachment model for anti-MPER antibodies. Interestingly, 2G12 was found to bind a fraction of unprocessed Env monomeric gp160 that incorporates into virions[30] and, as expected, we observe a higher degree of 2G12 binding when compared to anti-MPER antibodies (Supplementary Fig. 1b). Since the same distribution is detected for all tested bnAbs, our observations are further consistent with a clustering process mediated by the Gag-interacting Env tail[25,26], which appears to evolve independently of the Env quaternary structure.

### Colocalization of antibodies by dual color STED microscopy.

To further determine if anti-MPER antibodies bound to the same Env clusters as anti-gp120 bnAbs, the colocalization of the anti-MPER and anti-gp120 bnAbs signals was quantitatively assessed using dual color STED microscopy[31] (Fig. 3). In this approach, HIV-1 virions were incubated with anti-gp120 antibodies, which were detected with a secondary anti-human Fab antibody conjugated with the Abberior STAR 600 dye. After washing, these samples were subsequently treated with 10E8-KK114. Then, a pixel-wise Pearson's correlation test was run in viral signatures (Vpr.eGFP positive regions of interest, ROIs) to quantify the colocalization degree, according to which values of 1, 0, and −1 corresponded to maximal-colocalization, no-colocalization, and opposing-colocalization, respectively.

Figure 3a shows a representative image of the dual color STED microscopy experiment, where colocalization of the STED signals of VRC01 (blue) and 10E8 (red) antibodies can be observed in virus particles (green). To establish the significance of the correlation obtained, for each single experiment, image analyses were done as presented before for features on roundish objects[31]: (i) Vpr.eGFP positive areas (i.e. HIV-1 virions) (continuous line in the Fig. 3b top diagram); (ii) the same area after flipping one of the STED channels as a control for coincidental colocalization (i.e., colocalization that may arise from the restricted size of the analyzed area) (dashed line); and (iii) a random Vpr.eGFP-negative area (non-virus) (pointed line). Both, flipped and random controls showed correlation values close to 0, consistent with no-colocalization, which were significantly lower than those measured for the VRC01-10E8 sample (Fig. 3b, bottom).

Figure 3c displays colocalization data of three different anti-gp120 antibodies with 10E8. The broadly neutralizing VRC01 antibody recognizes the CD4-binding site (CD4bs) on gp120 protomers[32], whereas PGT145 bnAb binds specifically to the V1/V2 loops at the trimer apex[5,33] and, therefore, is thought to associate exclusively with properly folded Env (see also Fig. 1). In contrast to these antibodies, the antibody b12 was previously demonstrated to preferentially recognize and fix Env in a relaxed conformation[34].

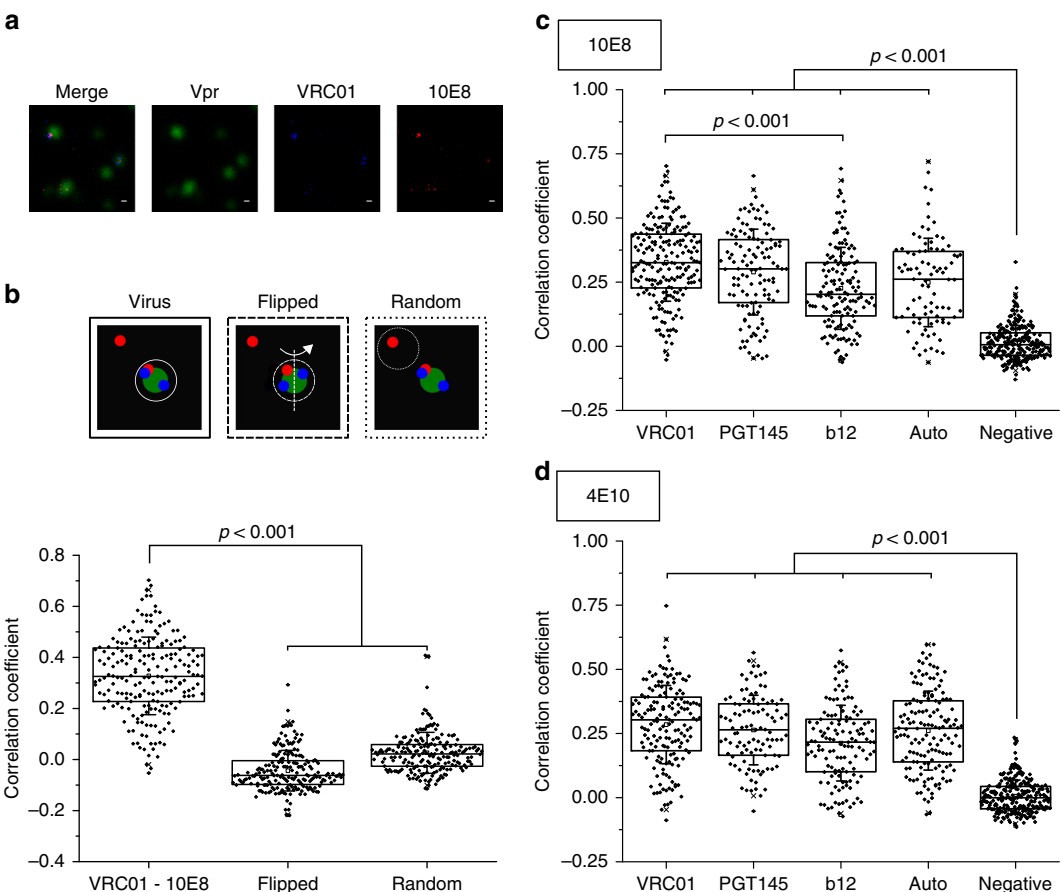

**Fig. 3** Dual color STED colocalization analysis of anti-gp120 and anti-MPER antibodies bound to Env$_{NL4-3}$ in HIV-1 virions. **a** Representative images of a colocalization experiment. Vpr.eGFP (green), VRC01 (STAR 600, blue, secondary labeling), 10E8 (KK114, red, direct labeling). Scale bars are 100 nm. **b** Example of colocalization analysis. Pearson's correlation coefficient for VRC01 and 10E8-KK114 ($n = 201$). Flipped: mirror image of one of the channels. Random: correlation in random ROIs negative for Vpr.eGFP. **c** Pearson's colocalization coefficients obtained for 10E8-KK114 in combination with STAR 600-labeled VRC01 ($n = 201$), PGT145 ($n = 120$), b12 ($n = 147$), and 10E8 itself (Auto) ($n = 89$). **d** Pearson's colocalization coefficients obtained for 4E10-KK114 (VRC01, $n = 156$; PGT145, $n = 103$; b12, $n = 130$ and 4E10 itself, $n = 148$). Conditions otherwise as in the previous panel. Results are shown in box-plots (center line, median; square, mean; box, IQR; whiskers, s.d.). The statistical significance was assessed by Kruskal–Wallis test

To establish maximal self-correlation values under these experimental conditions, dual color STED measurements of 10E8 were performed (i.e., colocalization analysis of 10E8-KK114, also stained with a secondary anti-Fab fluorescent STAR 600 conjugated antibody), which yielded median Pearson's coefficient of 0.26 (auto sample). As before, for all combinations tested, correlation values close to 0 were obtained for the flipped and random control measurements. In addition, samples incubated with fluorescently labeled secondary antibody in the absence of primary anti-gp120 antibody, were used as a negative control for colocalization with KK114-labeled 10E8 antibodies (negative). These samples, showing a median Pearson's value of 0.01, also confirmed the absence of cross-talk between the dual color detection channels. Finally, to obtain additional information on the dynamic range of our measurements, we established colocalization in a competitive binding experiment (Supplementary Fig. 2). As expected from the competitive binding of the two Abs, the pixel-wise correlation coefficient decreased significantly with respect to self-colocalization (Auto) controls.

As judged from the calculated correlation coefficients, 10E8 colocalized with the three anti-gp120 antibodies, the highest and lowest values being calculated for VRC01 and b12, respectively. A similar outcome was obtained for the 4E10-KK114 fluorescent conjugate (Fig. 3d). We note that STED does not offer enough resolution to distinguish whether these antibodies recognize the same molecule within Env clusters. However, maximal colocalization being attained with two of the most potent anti-gp120 antibodies described, i.e. VRC01, and the extremely trimer specific PGT145, strongly supports that anti-MPER antibodies are recognizing functional Env conformations relevant to HIV-1 infectivity.

**Accessibility to MPER established by quantitative imaging.** Even though 10E8 and 4E10 bind to an overlapping epitope, 4E10 is ca. 10–20 times less potent than 10E8 in terms of neutralization activity, and exhibits broad polyreactivity with lipids and host proteins[9,35,36]. The staining pattern of the fluorescently labeled 4E10 Fab bound to virions however reproduced 10E8's behavior regarding localization into defined foci, which were also stained by anti-gp120 antibodies (Figs. 2 and 3). Thus, 4E10's lower potency and higher polyreactivity appear not to alter the bnAb-binding pattern to Env complexes on NL4-3 intact virions. However, it has been described that in comparison with 10E8, access of 4E10 to MPER is more hindered in the neutralization resistant JR-CSF virions[9,37].

Given its potential to determine the number of single protein molecules at the nanoscale[38,39], we sought to establish differences in accessibility of these antibodies to native Env by quantitative STED imaging. Direct Fab labeling at 1:1 (Fab:probe) molar ratio and parallel sample preparation and imaging allowed direct comparison of KK114 intensities associated to individual virions (i.e. number of detected photons), further supported by the fact that different Fabs yielded the same photons/molecule values (Supplementary Fig. 3).

Comparison of independent sample replicates was performed after signal normalization to that of 10E8-KK114-treated NL4-3 virions. These measurements revealed comparable KK114 emission associated to NL4-3 virions treated with either 10E8 or 4E10 Fabs (Fig. 4a). To confirm the dependence of the KK114 emission levels measured on virions on specific Fab binding, we used the neutralization-deficient variants 10E8-W100bG (10E8-WG) and 4E10-Δloop produced by ablation of the HCDR3 loop tip[10,40,41]. Quantitation of number of detected photons revealed a strong reduction of binding in samples incubated with HCDR3 mutant conjugates 10E8-WG-KK114 or 4E10-Δloop-KK114, which

displayed antibody emission levels close to the background signal (Fig. 4a).

Quantitative analyses further supported the dependence of 10E8 and 4E10 binding to viral particles on the neutralization sensitivity of the Env glycoprotein (Fig. 4b–d and Supplementary Table 2). We compared binding of the KK114-labeled Fabs to three types of virions: (i) Env(-) particles lacking Env; (ii) neutralization-sensitive viruses containing $Env_{NL4-3}$; and (iii) neutralization-resistant, tier-2 viruses pseudotyped with $Env_{JR-CSF}$. Incubation of Env(-) particles with labeled Fabs resulted in background-like KK114 emission, thus further excluding the possibility of direct reactivity with the viral membrane. Overall, the amount of KK114 signal detected in association with the NL4-3 virions was higher than that detected in the JR-CSF samples, consistent with more hindered accessibility to MPER in the neutralization-resistant virions[9,37]. Of note, the similar values of mean virion trimer number ($\eta$) estimated for tier-1 ($\eta$, 13.5) and tier-2 ($\eta$, 11.8) viruses further suggest that this parameter does not account for the differences detected in the amount of Fab bound to viruses[42]. Moreover, the 10E8-KK114 signal was also higher than that of 4E10-KK114 in the JR-CSF virions.

STED microscopy[39] further allows the conversion of the intensity signals associated with virions (i.e., number of detected KK114 photons in eGFP-associated regions) into number of Fabs bound per viral particle ($F/v$ value). Supplementary Figure 3a displays the frequencies for the number of photons emitted by 10E8-KK114 and 4E10-KK114 bound to virions. After fitting emission histograms to a normal multipeak distribution (see Methods section), maxima separated by ca. the same number of photons ($\approx$50 in the example of Supplementary Fig. 3a) could be observed (Supplementary Fig. 3b). This value reflected the mean number of photons emitted by a single Fab bound to a virion in the sample. Furthermore, for each independent experiment, the value was the same within the experimental error for every tested fluorescent Fab, confirming a comparable quantum yield of the KK114 probe irrespectively of the specimen labeled (Supplementary Fig. 3b). Finally, further confirming the number of photons that come from a single labeled Fab, a single peak distribution with a maximum of $\approx$50 photons was also measured for both labeled Fabs stuck to the surface devoid of virus (Supplementary Fig. 3c).

Figure 4c, d compare the $F/v$ frequency distribution determined for 10E8 and 4E10 in single NL4-3 and JR-CSF virions, respectively. The most evident difference among the viral isolates is the slower $F/v$ decay exhibited by the neutralization-sensitive NL4-3 virions, in comparison with the neutralization-resistant JR-CSF virions, indicating higher degree of Fab accumulation in the neutralization-sensitive strain. Again, no major differences among Fabs were apparent in the NL4-3 sample. In the JR-CSF sample different $F/v$ decays were observed for the two Fabs. The steeper slope in 4E10 $F/v$ decay was in consonance with the majority of JR-CSF particles containing lower amounts of Fab 4E10 than of 10E8. Overall, virions presenting a single Fab molecule account for steep decays and correlate to lower neutralization potencies, suggesting that one Fab may not be enough to neutralize a virus.

Mean $F/v$ values calculated for all particles (i.e., including particles devoid of Fab) reflected comparable overall binding capacities for Fabs 10E8 and 4E10 in the laboratory-adapted strain NL4-3 (Supplementary Table 2). Mean $F/v$ values were also calculated only for particles displaying any KK114 signal (positive). These values ranged from ca. 4 to 6 in NL4-3 virions, upon incubation with 50 or 100 ng $\mu$L$^{-1}$ of Fab, respectively. By comparison, the $F/v$ values measured in the neutralization-resistant JR-CSF virus were generally reduced in the case of 4E10 (ca. 3.5), while the Fab 10E8 retained to a greater extent its

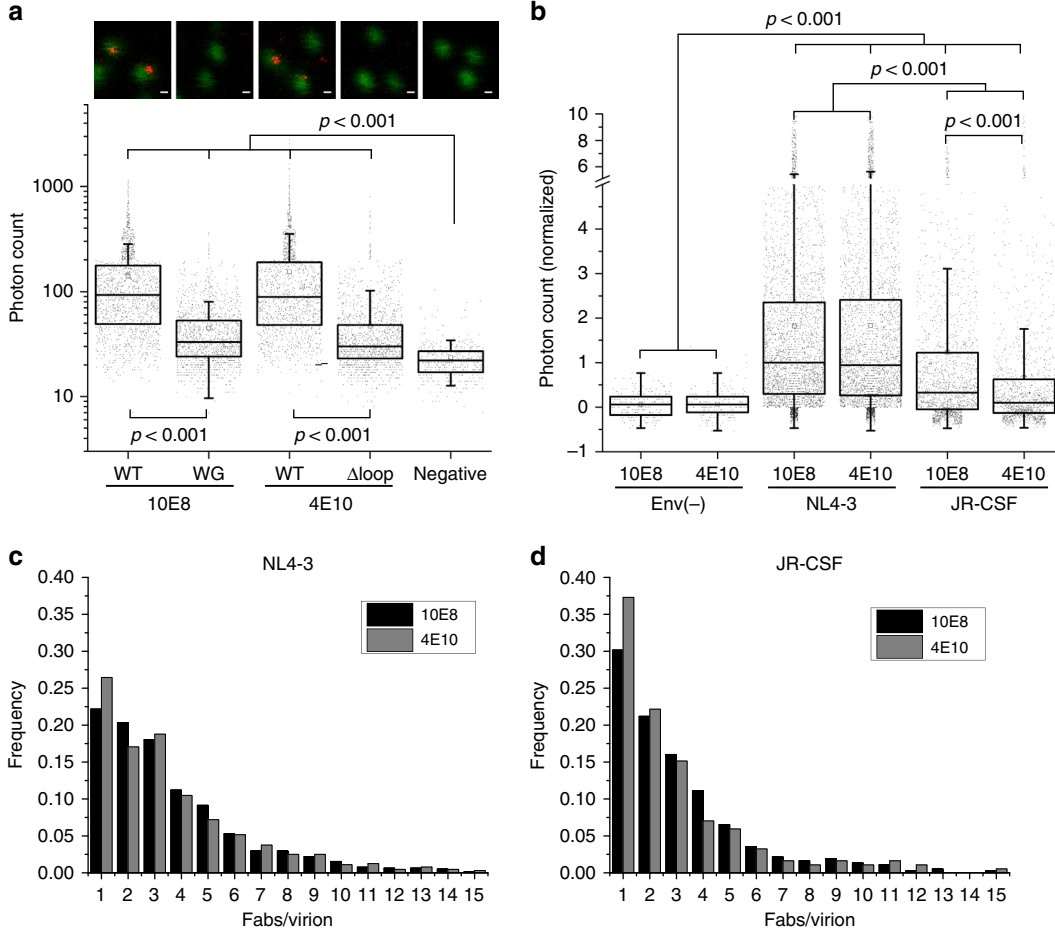

**Fig. 4** Differential binding of anti-MPER antibodies 10E8 and 4E10 to virions as revealed by quantitative STED microscopy. **a** Representative images of virions (Vpr.GFP, green) and antibodies (KK114, red) (top) and measured KK114 intensities (bottom) for 10E8 WT ($n = 776$), 10E8 WG ($n = 1375$), 4E10 WT ($n = 644$), and 4E10 Δloop ($n = 1109$). Each point in the bottom plot represents the KK114 intensity sum measured in a single viral particle. Negative samples ($n = 795$) were measured in the absence of labeled antibody. Data from a single experiment. **b** 10E8 and 4E10 STED signals measured in Env(-) ($n = 223$ and 202), Env$_{NL4-3}$ ($n = 3336$ and 3165), and Env$_{JR-CSF}$ ($n = 1664$ and 1303) HIV-1 virions. For the matter of comparison signals from single virions from two or more independent experiments have been normalized to 10E8-Env$_{NL4-3}$ signal after background subtraction. Histograms displaying $F/v$ frequencies measured for both antibodies (50 ng μL$^{-1}$) in NL4-3 (**c**) and JR-CSF (**d**) viruses. The statistical significance was assessed by Kruskal–Wallis one-way analysis of variance. If not noted otherwise, differences were not significant at the 0.05 level. In **a** and **b** results are shown in box-plots (center line, median; square, mean; box, IQR; whiskers, upper and lower inner fences). Scale bars are 100 nm

capacity for binding (4.3 and 5.4 for 50 and 100 ng μL$^{-1}$, respectively). In conclusion, the amount of Fab detected in association with the NL4-3 virions was higher than that detected in the JR-CSF samples, thus following their neutralization-sensitivity pattern. Moreover, in the case of the tier-2 virions, higher amounts of 10E8 than of 4E10 were bound to the samples, again in agreement with a better adaptation of the former antibody to access the native MPER.

**Contribution of 10E8 structural determinants to binding.** Structural, biophysical, and biochemical data[9–11,15,43,44] suggest that the bnAb 10E8 employs three elements to facilitate access to and recognition of the membrane-inserted MPER (Fig. 5a): (1) a specificity-binding pocket, which is adapted to recognize a MPER α-helical structure, and forms a hydrophobic cleft where residues critical for binding Trp$_{672}$ and Phe$_{673}$ are buried[9,10,45]; (2) a long heavy chain complementarity determining region 3 (HCDR3) loop, characterized by a very hydrophobic tip that submerges into the lipid matrix[9–11,43]; and (3) the membrane-associated paratope area (MAPA) that can accommodate phospholipid polar head groups on its surface[10,11,15,43].

To establish the influence of these elements on the antibody's capacity to access the native MPER, Env and Fab 10E8 variants were produced, and antibody binding to JR-CSF virions was subsequently evaluated through quantitative STED microscopy (Fig. 5b–d). The dependence of 10E8 recognition on Env residues Trp$_{672}$ and Phe$_{673}$ was first tested using particles that contained the gp41 subunit mutated in the MPER epitope (Fig. 5b). Confirming binding specificity, single F673A or double W672A/F673A mutations of MPER residues severely limited, or totally inhibited antibody binding to the viral particles. Several 10E8 variants were next generated to evaluate the contribution of antibody–membrane interactions to Env binding (Fig. 5c, d and Supplementary Fig. 4). Fab modifications combined deletions at the tip of the HCDR3 loop (WG mutant) with substitutions of solvent-exposed MAPA residues (marked blue in Fig. 5a) by Arg to augment electrostatic attraction to the negatively charged membrane surface (3R mutants[44]). Lipid vesicle flotation assays displayed in Supplementary Figure 4a illustrate the distinct degrees of 10E8–membrane interactions induced by these modifications. Consistent with previous reports[10,21], the WT version of 10E8 displayed no spontaneous binding to membranes,

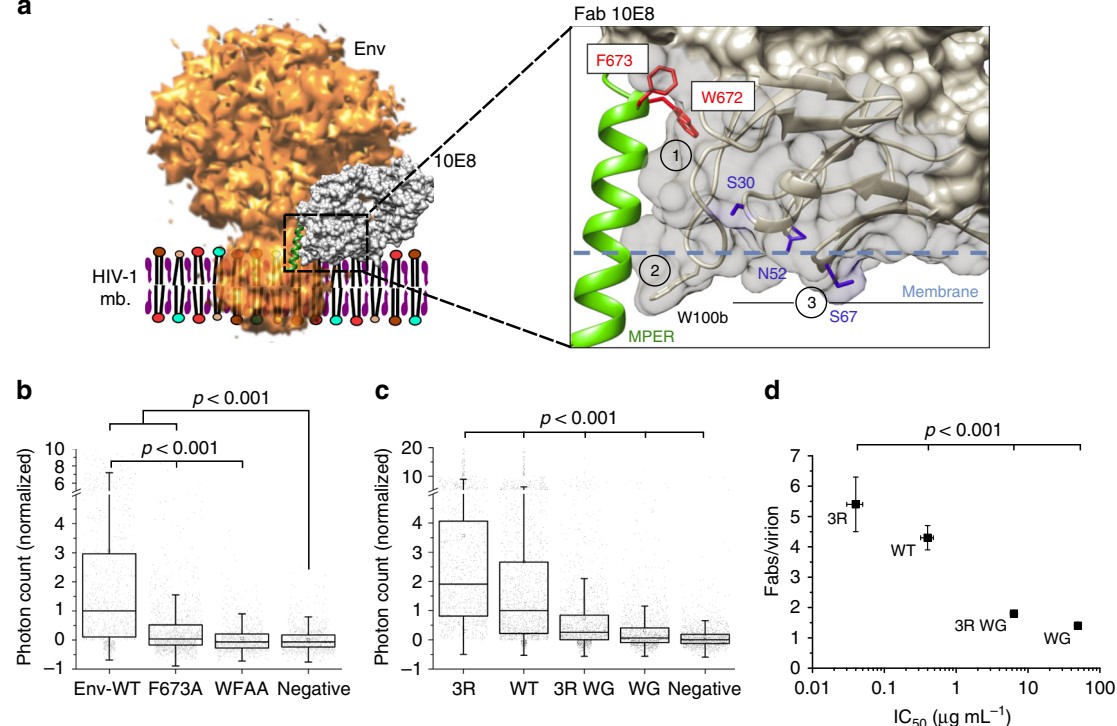

**Fig. 5** Dependence of 10E8 binding to native MPER on the functional paratope elements. **a** Structural model to designate: (1) specificity pocket, (2) HCDR3, and (3) MAPA regions on 10E8 (EMD entry code 3308[15] and PDB 5GHW[10]). Side chains of residues $Trp_{672}$ and $Phe_{673}$ critical for epitope peptide binding and neutralization are displayed in red onto the bound MPER helix (green ribbon). Fab residues from the HCDR3/MAPA region modulating membrane interaction and neutralization activity, $Trp_{100bHC}$, and residues mutated to Arg (in blue: $Ser_{30LC}$, $Asn_{52LC}$, $Ser_{67LC}$) are displayed in stick representation. **b** STED intensity signals of the antibodies (after background subtraction and normalization) in HIV-1 virions pseudotyped with JR-CSF Env-WT ($n = 939$) and Env versions with single F673A ($n = 1270$) or double W672A/F673A (WFAA) ($n = 884$) mutations in the MPER. **c** Binding to Env on intact virions of anti-MPER bnAbs mutated in the MAPA/HCDR3 region. The intensity signals for each antibody was normalized to the WT signal after background subtraction (3R $n = 1049$, WT $n = 1664$, 3R WG $n = 1437$, WG $n = 1444$). **d** Correlation between potency ($IC_{50}$ values for neutralization, determined by cell entry inhibition assay) and mean $F/v$ values determined by quantitative STED in the previous samples. The statistical significance was assessed by Kruskal–Wallis test. If not noted otherwise, differences were not significant at the 0.05 level. Results are shown in box-plots (center line, median; square, mean; box, IQR; whiskers, upper and lower inner fences)

a pattern reproduced by the WG loop mutant. Interestingly this mutant gained capacity for associating with model lipid vesicles upon inclusion of the 3R mutation (3R/WG mutant, exposing basic Arg residues on the MAPA). The 3R mutant with an intact HCDR3, also showed capacity for membrane partitioning. Thus, membrane interactions appear to be promoted by strengthening electrostatic interactions through the MAPA, even after suppressing hydrophobicity at the HCDR3 tip.

Figure 5c, d, and Supplementary Figure 4b compare binding of these 10E8 Fab variants to intact JR-CSF virions as measured by quantitative STED microscopy imaging. The WG mutation of the hydrophobic HCDR3 tip greatly reduced KK114 intensity associated with virions (Fig. 5c), indicating a lower $F/v$ value for this sample (Supplementary Figure 4b). Partial recovery of Fab binding was observed for the 3R-WG mutant. The 3R mutant with intact HCDR3 displayed KK114 intensity levels and $F/v$ values that were higher than those of the WT Fab. The plot in Fig. 5d displays the correlation existing between the neutralization potency of the mutants ($IC_{50}$ values) and the $F/v$ values determined through STED imaging.

Notably, we also observed that, following the pattern described for the parental antibody, the most potent antibody 10E8 3R did not interact with membrane zones devoid of glycoprotein or Env (-) virions, and associated to Env clusters (Supplementary Fig. 5). We emphasize that the absence of interaction with the bare viral membrane detected by STED microscopy is not at odds with

previous analyses demonstrating that 10E8 3R and 4E10 antibodies can interact directly with model membranes[16,40,44,46]. Membrane-association has been customarily established using lipid compositions that facilitate MAPA-mediated electrostatic interactions (e.g., the PC-based mixtures containing high levels of PS in Supplementary Fig. 4a). However, in the light of recent observations that describe the functional organization of the HIV membrane[47], it has become evident that the lipid mixtures favoring spontaneous bnAb partitioning into membranes do not reflect the more restrictive conditions imposed by the highly ordered viral membrane. Consistently, Fabs 10E8 3R or 4E10 were unable to interact directly with vesicles made of a virus-like mixture, which displays lipid packing levels comparable to those of the viral membrane[47] (Supplementary Figs. 4a and 6, bottom rows). Overall, these data support that mutations, such as 3R, which have a positive effect on the neutralization potency and accessibility to the MPER epitope, enhance membrane interactions of the Fab 10E8 after specific recognition of Env.

## Discussion

Induction of bnAbs that inactivate and clear HIV is an important objective of a preventive HIV vaccine[48,49]. In this regard, establishing Env antigenic structures responsible for the elicitation of bnAbs is critical for vaccine design and immunotherapy[5,50]. The highly conserved MPER constitutes a prime target, since

antibodies reported to bind this site of vulnerability display broad coverage and in vivo anti-viral activity[51–55]. However, the structural context where bnAbs effectively engage the native MPER remains largely undefined, in great part because of the difficulties associated with establishing adequate model systems that mimic the viral membrane-anchored Env complex (Fig. 1).

To circumvent that problem, here we have approached the question of MPER accessibility on native Env by directly imaging the interactions of fluorescent antibodies with intact virions using STED microscopy. This super-resolution microscopy technique can resolve details within sub-diffraction-sized HIV-1 particles and detect the signal of a single fluorescent Fab molecule bound to a virion. Moreover, the fact that our specimens were directly labeled with KK114 at ca. 1:1, Fab-to-probe molar ratio, has permitted the quantification of emitted photons per molecule and the estimation of the number of antibodies bound to a single virion, further allowing the comparison of the binding potency of different antibody variants. Overall, STED microscopy revealed two important sets of data, namely: (i) the occurrence of similar patterns of native Env recognition for anti-MPER antibodies and for antibodies against the solvent-exposed surface subunit gp120 (Figs. 2 and 3); and (ii) a correlation between the anti-viral activity of anti-MPER antibodies and their ability to engage native Env on virions (Figs. 4 and 5). We propose that these observations have important implications for the understanding of the mechanisms that underlie elicitation and anti-viral activity of antibodies targeting the MPER site of vulnerability.

First, our data are compatible with a model in which MPER, accessible in the native Env, mediates initial engagement of anti-MPER bnAbs with virions. Based on the high hydrophobicity of their HCDR3 element, it had been suggested that anti-MPER bnAbs attach first to the viral membrane (pre-attachment model), where they reside as integral membrane proteins until engagement with their MPER epitope. MPER would become only transiently accessible on the pre-hairpin fusion intermediate of Env[5,16–19,21]. Data displayed in Figs. 2 and 4 indicate that the anti-MPER bnAbs recognize Env clusters, are absent from membrane zones devoid of glycoprotein, and do not interact with Env(-) particles. Moreover, both 10E8 and 4E10 colocalize with bnAbs targeting the solvent-exposed gp120 subunit (Fig. 3). In combination, these observations are inconsistent with the existence of an anti-MPER antibody population bound to the viral membrane (i.e. disconnected from Env) waiting for MPER to be accessible in HIV-1 virions.

Second, it appears that the degree of native MPER accessibility correlates with the anti-viral activity of the antibodies. Thus, following the neutralization sensitivity, higher Fab/virion values were measured for $Env_{NL4-3}$ compared to $Env_{JR-CSF}$ virions (Fig. 4). Also, in accordance with previously reported experiments washing antibody–virion mixtures prior to infection[9,37], the amount of 10E8 bound to tier-2 JR-CSF virions was higher in comparison with that measured for 4E10. These data suggest that 10E8 is better adapted than 4E10 to access sterically occluded native MPER. Supporting different mechanisms of Env recognition, structural data of Fab-peptide complexes reveal different docking angles for 10E8 and 4E10[10,11,56]. In addition, 10E8 establishes a firm grip along the full MPER helix, including the interaction between $Trp100b_{HC}$ and TMD residues, $Ile_{686}$ and $Met_{687}$[10] (Fig. 5a). Wrapping of the accessible helical surface by the HCDR3 of the 10E8 appears energetically favored at the membrane interface by an aromatic collar of antibody and MPER residues[10]. In comparison, X-ray crystallography reveals a smaller surface of the MPER helix buried into the 4E10-binding pocket, whereas its HCDR3 tip makes no contact with the bound peptide[57]. Nonetheless, which bnAb attribute is responsible for the better adaptation of 10E8, if any, still remains to be fully elucidated.

The existence of an accessible MPER helix within the viral membrane-anchored native Env complex is further supported by quantitative imaging, which reveals that an increase in the potency of 10E8–membrane interactions through MAPA/HCDR3 translates into enhanced binding to virions and improved antiviral efficacy (Fig. 5). Altogether, these results suggest that structural adaptations that sustain anti-MPER antibody–lipid interactions and increase neutralization potency, have arisen to enhance affinity toward an MPER helix already accessible within the membrane-anchored pre-fusion Env complex.

In the recent years, we have witnessed a rapid development of super-resolution microscopy and its application to the HIV field (reviewed in ref. [58]). STED is of special interest because it can be combined with advanced microscopy methods, such as molecular dynamic measurements by fluorescence correlation spectroscopy (STED-FCS) or membrane molecular order measurements using polarity sensitive dyes (STED-GP), as applied for HIV in refs. [26,59]. Future work will aim to combine these advanced STED approaches with live-cell imaging and monitor HIV cell entry and its neutralization by bnAbs. Thus, measurements of Env, lipid, and antibody mobility during the different entry steps (CD4 receptor and CXCR4/CCR5 co-receptor binding, Env unfolding and membrane fusion) will reveal the role of each player in bnAb-mediated neutralization and the dynamics of the process. Emphasizing the possible relevance of Env activation in the process, previous observations by electron microscopy suggest that 4E10 binding to virions can be accelerated in the presence of soluble CD4[60]. Moreover, the interaction potency of bnAbs and optimized versions (as the 3R variant presented in this work) would be measured in situ, further permitting the identification of the Env structures recognized and the entry steps inhibited by anti-Env antibodies. These experiments will inform approaches to design anti-Env vaccines and clarify what elements of anti-Env antibodies can be subject to optimization when used as templates for immunotherapeutic agent development[61].

## Methods

**Plasmid and cells.** The pCHIV plasmid expressing all HIV-1 proteins except Nef and lacking the viral long-terminal repeat sequences was used to produce replication-incompetent HIV-1$_{NL4-3}$ virions. A derivative of this plasmid including a premature termination of Env was used to produce Env(-) particles[62]. HIV-1 expression plasmid pCHIV and its derivative were provided by Barbara Müller and Hans-Georg Kräusslich (University Hospital, Heidelberg, Germany). Alternatively, HIV-1$_{JR-CSF}$ pseudoviruses were produced upon transfection of the Env(-) pCHIV plasmid and the full-length JR-CSF Env clone. Plasmid expressing eGFP.Vpr was kindly provided by Tom Hope. 293T cells (ATCC CRL-3216) were grown in Dulbecco's modified Eagle's medium (Sigma), supplemented with 10% fetal calf serum, 100 U mL$^{-1}$ penicillin–streptomycin and 20 mM HEPES pH 7.4. Cells were maintained at 37 °C, 5% $CO_2$.

**Antibodies and dyes.** The sequences of 10E8 or 4E10 were cloned in the plasmid pColaDuet and expressed in *Escherichia coli* T7-shuffle strain. Recombinant expression was induced at 18 °C overnight with 0.4 mM isopropyl-D-thiogalacto-pyranoside when the culture reached an optical density of 0.8. Cells were harvested and centrifuged at 8000 × g, after which they were resuspended in a buffer containing 50 mM HEPES (pH 7.5), 500 mM NaCl, 35 mM imidazole, DNase (Sigma-Aldrich, St. Louis, MO), and an EDTA-free protease inhibitor mixture (Roche, Spain). Cell lysis was performed using an Avestin Emulsiflex C5 homogenizer. Cell debris was removed by centrifugation, and the supernatant loaded onto a nickel-nitrilotriacetic acid (Ni-NTA) affinity column (GE Healthcare). Elution was performed with 500 mM imidazole, and the fractions containing the His-tagged proteins were pooled, concentrated, and dialyzed against 50 mM sodium phosphate (pH 8.0), 300 mM NaCl, 1 mM DTT, and 0.3 mM EDTA in the presence of purified protease Tobacco etch virus. Fabs were separated from the TEV and cleaved peptides containing the His6× tag by an additional step in a Ni-NTA column. The flow-through fraction containing the antibody was dialyzed overnight at 4 °C against sodium acetate (pH 5.6) supplemented with 10% glycerol and subsequently loaded onto a MonoS ion exchange chromatography (IEC) column (GE Healthcare). Elution was carried out with a gradient of potassium chloride and the fractions containing the purified Fab concentrated and dialyzed against a buffer

containing 10 mM sodium phosphate (pH 7.5), 150 mM NaCl, and 10% glycerol. For the preparation of mutant Fabs, the KOD-Plus mutagenesis kit (Toyobo, Osaka, Japan) was employed following the instructions of the manufacturer. Anti-MPER Fab labeling was attained by introducing first titratable Cys residues at positions $Cys_{216HC}$ and $Cys_{228HC}$ of the 10E8 and 4E10 sequences, respectively, and then by modifying those with a sulfhydryl-specific iodacetamide derivative of the Abberior STAR RED (KK114) probe (Abberior GmbH, Göttingen, Germany). After purification, fluorescence emission measured after SDS–PAGE and absorbance measurements confirmed almost total titration of the single free Cys residues in the Fabs. Human anti-gp120 monoclonal antibodies 2G12 and b12 were purchased from Polymun Scientific. Fab fragments were generated using the Fab Micro Preparation kit (Pierce). Anti-human IgG Fab fragments (Jackson ImmunoResearch) were coupled to the Abberior STAR 600 dye (Abberior GmbH, Göttingen, Germany) via NHS-ester chemistry according to the dye manufacturer's instructions. The heavy chain and light chain of PGT145 and VRC01 Fabs were each synthesized by GeneArt (Life Technologies) and subcloned into the pHLsec mammalian expression vector using restriction enzymes AgeI and KpnI[63]. The heavy chain and light chain of each Fab were co-transfected into HEK293F cells in a 2:1 ratio (90 μg of DNA total per 200 mL culture) using the FectoPRO transfection reagent (Polyplus Transfections) (1:1 ratio of DNA:FectoPRO) at a cell density of $0.8 \times 10^6$ cells mL$^{-1}$. Cultures were incubated in a Multitron Pro shaker (Infors HT) at 37 °C, 125 rpm, 70% humidity, and 8% CO$_2$. One week later, cells were harvested at 6000×g for 30 min and resulting supernatants were filtered with a 0.22 μm filtration device (EMD Millipore). Supernatants were loaded onto a pre-equilibrated KappaSelect affinity column (GE Healthcare) at a flow rate of 4 mL min$^{-1}$. The column was washed with $1 \times$ PBS (~3 column volumes) and Fab fragments were eluted with 100 mM glycine pH 2.2 while fractionating. Eluted fractions were neutralized immediately with 10% (v/v) 1 M Tris–HCl, pH 9.0 and were pooled and concentrated to run on a Superdex 200 Increase gel filtration column (GE Healthcare) in $1 \times$ PBS to obtain purified samples. Peaks were pooled for downstream experiments. Sample purity was confirmed by SDS–PAGE.

**Virus particle preparation and purification.** Virus particles were prepared as previously described[26]. Briefly, 293T cells were transfected using polyethylenimine; tissue culture supernatants were harvested 48 h after transfection, cleared by filtration through a 0.45 μm nitrocellulose filter, and particles were purified by ultracentrifugation through 20% (w/v) sucrose cushion at 70,000 × g (avg.) for 2 h at 4 °C. Particles were resuspended in ice-cold 20 mM HEPES/PBS pH 7.4, snap frozen and stored in aliquots at −80 °C. All ultracentrifugation steps were performed in a SW 41 Ti rotor.

**Microscopy sample preparation.** Purified virus particles were adhered to poly-L-lysine (Sigma)-coated glass coverslips for 15 min. Coverslips were blocked using 2% fatty acid free bovine serum albumin (BSA) (Sigma)/PBS for 15 min. Abberior STAR RED (KK114) conjugated anti-MPER Fabs (20–100 ng μL$^{-1}$) were incubated for 1 h in blocking buffer. Annexin V was incubated in 10 mM CaCl$_2$/HBS. Immunostained particles were washed and mounted in PBS, followed by STED analysis. All steps were carried out at room temperature. For colocalization experiments, before adding labeled anti-MPER Fabs (100 ng μL$^{-1}$), unlabeled anti-Env Fabs (1 h, 100 ng μL$^{-1}$ in blocking buffer) and anti-human Abberior STAR 600 conjugated Fab fragments (1 h, 1:100 in blocking buffer) were incubated. Three very gentle 5-min PBS washing steps were performed before and after adding the secondary Fabs. Control experiments without anti-Env primary antibodies show no Abberior STAR 600 signal, indicating that washing steps successfully removed free anti-human secondary Fabs from the sample.

**STED microscopy measurements.** Imaging was performed on a STED microscope based on a modified Abberior Instrument RESOLFT QUAD-P super-resolution microscope (Abberior Instruments GmbH) installed in a biosafety level 3 environment. The microscope was equipped with three pulsed excitation lasers (485, 594, and 640 nm; LDH-D-C-485P and LDH-D-C-640P, Picoquant, Berlin, Germany, and LightUp594, Abberior Instruments) with 80 ps pulse width and a pulsed STED laser (Katana HP, Onefive GmbH, Switzerland) operating at 775 nm, 800 ps pulse width and 80 MHz repetition rate. Shuttering and power adjustment of the STED laser were controlled with an acousto-optical modulator (MT110-B50A1.5-IRHK, AAA/Photon Lines Ltd, Banbury, UK). Doughnut shaped focal intensity distribution of STED laser was achieved by easy3D STED spatial light modulator module (Abberior Instruments). STED and excitation laser beams were spatially superimposed and the fluorescence light was separated using appropriate dichroic filters (ZT740SPRDC, AHF Analysentechnik, Tübingen, Germany). An on-board FPGA card (Abberior Instruments) was used for time alignment control between the laser pulses. Positioning and scanning of laser foci was realized using the QUAD beam scanner unit of the Abberior system for lateral directions, and an objective lens positioning system (MIPOS 100PL, Piezosystem Jena, Jena, Germany) for the axial direction. The fluorescence excitation and collection was performed using a ×100/1.40 NA UPlanSApo oil immersion objective (Olympus Industrial, Southend-on-Sea, UK). The fluorescence signal was descanned, passed through an adjustable pinhole (Thorlabs Limited, Ely, UK) and detected by a single photon counting avalanche photo diode (SPCM-AQRH-13, Excelitas Techologies)

with appropriate fluorescence filters (AHF Analysentechnik). Detected fluorescent signal was time-gated to remove the fluorescent signal from undepleted fluorophores when operating in STED mode. All acquisition operations were controlled by Inspector software (Abberior Instruments). In dual color STED microscopy measurements both signals were recorded line by line in STED microscopy mode, then whereas eGFP.Vpr signal was imaged in confocal mode to determine the location of HIV-1 virus particles. The lateral spatial resolution was below 60 nm FWHM in both STED acquisition channels. Following parameters were used during image acquisition: pinhole size – 1 airy unit, pixel dwell time – 50 μs, field of view −10 μm × 10 μm, and pixel size – 20 nm.

**Image analysis.** Image analysis was performed using Python scripting language and custom written functions based on the program developed for ref. [31]. Individual viral particles were identified from the Vpr.eGFP channel using an intensity maximum finding algorithm on a Gaussian smoothed image ($\sigma = 2.0$). Detection of maxima was kept consistent throughout using a noise tolerance parameter of 10. A circular region (diameter, 20 pixels; 400 nm) was then superimposed on each detected location, and all of the regions were saved for subsequent analysis. For every detected region, a random location was also generated to sample areas where Vpr.eGFP staining and thus HIV-1 virions were not likely to be present. This was achieved by randomly translating each of the detected regions to a different point within a 90-pixel radius of the original location but constrained so as not to pick an existing region, which might contain another fragment of Vpr.eGFP fluorescence. This method was effective at finding random regions that were close to virions but not overlapping and so ensured accurate comparisons between virion-containing and non-virion regions. These randomly perturbed regions were saved and used for subsequent comparisons as with the original set.

Correlation analysis was performed on the raw pixel data in each of the previously detected regions (Fig. 3). The intensity contained within each region was first integrated, in the Vpr.eGFP confocal and in the STED channel under comparison, to form datapoints ($g_i, r_i$, respectively), and then Pearson's correlation coefficient was calculated from all the measurements ($I$) in each cell.

$$\rho = \frac{\sum_{i=1}^{I} \left(g_i - \mu_g\right)\left(r_i - \mu_r\right)}{\sqrt{\sum_{i=1}^{I} \left(g_i - \mu_g\right)^2 \sum_{i=1}^{I} \left(r_i - \mu_r\right)^2}} \quad (1)$$

where $\mu_g$ and $\mu_r$ are equal to the mean intensity of all the summed regions in the sample. Since not all the virions present both or even one type of antibody, to ensure that the colocalization values were not underestimated, only virions presenting a minimum of one focus in both STED channels (i.e., virions positive for both antibodies) were selected for the analysis. It must be noted that no cooperativity between any of the tested anti-gp120 antibodies and 10E8 was observed, since we detected the expected frequency of double positive, i.e. the product of the frequency of positive virions for each type of bnAb.

To determine the number of fluorescent Fab molecules bound to each individual virus ($F/v$ value), the number of emitted photons per Fab-KK114 molecule was estimated first. For that, frequency histograms of KK114 photons detected in virions (i.e. Vpr.eGFP positive areas) positive for KK114 signal (Supplementary Fig. 1a) were fit to a multi-peak Gaussian distribution. The peak maximum values calculated from these fittings were divided by the corresponding natural numbers and averaged to obtain the photons/molecule value (Supplementary Fig. 1b), which were the same within the experimental error for samples prepared and measured in parallel. The result was confirmed by measuring Fab-KK114 molecules bound to the coverslip. To calculate the $F/v$ value, photons detected in Vpr.eGFP areas were divided by the photons/molecule value. To calculate the area occupied by the antibodies, the threshold is calculated by applying a Renyi entropy algorithm[64] for each patch region. Then it is denoised with a $3 \times 3$ median filter and eroded by one pixel.

**Cell entry inhibition.** HIV-1 pseudoviruses were first produced by transfection of human kidney HEK293T cells (ATTC CRL-3216) with the full-length env clones HXB2 and JR-CSF (kindly provided by Jamie K. Scott and Naveed Gulzar, Simon Fraser University, Burnaby, Canada) using calcium phosphate. The cells were co-transfected with vectors pWXLP-GFP and pCMV8.91, encoding a green fluorescent protein and an env-deficient HIV-1 genome, respectively (generously provided by Patricia Villace, CSIC, Madrid, Spain). After 24 h, the medium was replaced with Optimem-GlutaMAX II (Invitrogen) without serum. Three days after transfection, the pseudovirus particles were harvested, passed through 0.45 μm pore sterile filters (Millex HV; Millipore NV, Brussels, Belgium), and finally concentrated by ultracentrifugation in a sucrose gradient. Neutralization was determined using TZM-bl target cells (AIDS Research and Reference Reagent Program, Division of AIDS, NIAID, NIH, contributed by J. Kappes) as described previously[65]. Samples were set up in duplicate in 96-well plates and incubated for 1.5 h at 37 °C with a 10–15% tissue culture infectious dose of pseudovirus. After antibody-pseudovirus coincubation, 11,000 target cells were added in the presence of 30 μg mL$^{-1}$ DEAE-dextran (Sigma-Aldrich). Neutralization levels after 72 h were inferred from the reduction in the number of GFP-positive cells as determined by flow cytometry using a BD-FACSCalibur flow cytometer (Becton Dickinson).

**Statistical analysis**. All statistical analyses were performed using SPSS. Comparison between multiple groups were done using the nonparametric Kruskal–Wallis test, which does not assume a normal distribution. An adjusted $p <$ 0.05 was considered significant.

**Code availability**. The image analysis code DOI is 10.5281/zenodo.1465920.

## Data availability
The data that support the findings of this study are available from the corresponding authors upon request.

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

## Acknowledgements
This study was supported by the Spanish MINECO (BIO2015-64421-R (MINECO/FEDER UE) to J.L.N.) and the Basque Government (IT838-13 to J.L.N.). P.C., E.R., and S.I. received pre-doctoral fellowships from the Basque Government. P.C. would like to acknowledge the European Biophysical Societies' Association (EBSA) for receiving an EBSA Bursary for a working visit to a laboratory in an EBSA country. J.C., D.W., and C.E. greatly acknowledge support by the MRC (grant number MC_UU_12010/unit programs G0902418 and MC_UU_12025), the Wellcome Trust (grant 104924/14/Z/14 and Strategic Award 091911 (Micron)), MRC/BBSRC/EPSRC (grant MR/K01577X/1), BBSRC (Deutsche Forschungsgemeinschaft (Research unit 1905 "Structure and function of the peroxisomal translocon")), the Wolfson Foundation (for initial funding of the Wolfson Imaging Centre Oxford), the EPA Cephalosporin Fund and the John Fell Fund. T.S. is a recipient of a Canada Graduate Scholarship Master's Award and a Vanier Canada Graduate Scholarship from the Canadian Institutes of Health Research. This work was supported by operating grant NIH-150414 (J.-P.J.) from the Canadian Institutes of Health Research. This research was undertaken, in part, thanks to funding from the Canada Research Chairs program (J.-P.J.). We acknowledge valuable technical assistance from Miguel García-Porras.

## Author contributions
P.C., J.C., C.E., and J.L.N. designed research; P.C., J.C., E.R., S.I., B.A., and T.S. performed research; D.W. and E.L. contributed analytic tools; P.C. and J.L.N. analyzed data; and P.C., J.C., E.R., J.-P.J., C.E., and J.L.N. wrote the paper with input from all authors.

## Additional information

**Competing interests:** The authors declare no competing interests.

