## [Peer Review File · Nature Communications]

Reviewers' comments:

Reviewer #1 (Remarks to the Author):

Caravilla P. et al. presented a study of the accessibility of antibodies to the native Env MPER through stimulated-emission-depletion (STED) microscopy.

To my opinion, the importance of this study can be approached from two different perspectives.

The first is related to the results, the authors clearly provide new biophysical insights into the recognition of the potent and broadly neutralizing MPER epitope on HIV virions.

The second is related to a good example of the power of STED microscopy and the importance of solid labelling protocols when using such technique - more in general when using super-resolution microscopy. Regarding the second topic I am happy and impressed to see that this is one of the first studies that use STED microscopy imaging in a very quantitative manner - STED-FCS is quantitative by nature. In particular, the authors took advantage of a very important - but sometimes underestimated - property of STED microscopy (preserved also in its time-gated implementation), namely the linearity of the intensity signal with the concentration of the labelled molecules/proteins. This property is not always maintained in many emerging point-scanning microscopy techniques, which heavily process the data/signal provided by the microscope. Here, authors can effectively count the number of Fabs bound per viral particle.

For these reasons I fully support the publication of this study on Nature Communications.

Minor points:

- I was not able to find details about the image segmentation procedure implemented to measure the Area (in pixel) for Fig. 2 B. Please, provide it.

- I suppose that the images are collected with a gated pulsed STED implementation. Since the manuscript highlights the methodology aspect of this study, and in particular the STED microscopy method, I would like to read a little bit more details of the STED implementations.

Suggestions:

- Since the authors show a very multi-disciplinary aspect, from microscopy to image analysis, to biophysics, I will be happy to read in the conclusion a comment regarding what would be the next important steps from STED microscopy in the biological application addressed in this work.

Reviewer #2 (Remarks to the Author):

Carravilla et al. have reported a study of anti-MPER Ab binding to Env spikes on the virion surface using STED microscopy. They demonstrate that anti-MPER Ab association with the virion is entirely dependent on the presence of Env, arguing against any interaction with the viral membrane that might precede interaction with Env. Instead they argue in favor of the MPER region being intrinsically exposed due to the inherently dynamic nature of Env. They go on to identify a correlative relationship between the number of Fab fragments bound per virion and neutralization potency. This study provides a nice example of how contemporary quantitative fluorescence microscopy can provide structural information under circumstances not amenable to conventional structural determination.

1. The authors have provided no explicit indication of whether the Fab binding sites are fully saturated during preparation of their samples. This is clearly important to ensure that their estimations of the number of Env clusters per virion are reliable. Showing that a higher concentration of Ab yields the same result would be helpful in this regard.

2. The authors demonstrate that a panel of Abs that target either the MPER or diverse epitopes in gp120 all yield the same apparent distribution of Env. This would not be surprising if all the Abs tested were specific to the native tertiary structure of the trimeric Env. However, 2G12 is a relatively weakly neutralizing Ab in the present context, and certainly not specific to the native Env trimer. The Binley laboratory has shown that HIV-1 virions have an abundance of non-functional Envs, including uncleaved Env precursors, monomers, dimers, post-fusion gp41 stumps, and so on. Therefore, one might expect that 2G12 would indicate a distinct Env distribution, or perhaps a greater number of visible (Ab-bound) Envs per virion. This should be discussed.

3. How does colocalizing anti-MPER Abs with another Ab indicate “neutralization relevant binding events”? This is not clearly described.

4. Why are the correlation coefficients not higher? Obviously in the presence of noise, one would not observe perfect correlation. But could this also reflect limited binding efficiencies? Or competitive binding of the two Abs? For example, a single PGT145 binds the trimer apex (Lee et al. Immunity 2017). Can it bind to an open conformation with the MPER exposed? Interpretation of these results would be helped by measuring the pixel-wise correlation coefficient for Abs with overlapping epitopes (ie, competitive Abs). This would indicate their full dynamic range in determination of the correlation coefficient.

5. For the measurements of the number of Fabs per virion, the photon count distributions need to be calibrated using individual Fabs stuck to the surface in the absence of virus, similarly to that done by Chojnacki et al. (Science 2012). Without this control, they have no way of verifying the number of photons that come from a single labeled Fab.

6. The Trkola laboratory has reported that NL4-3 has somewhat more Env trimers per virion than JR-FL (Brandenberg et al. PLoS Path 2015). While the difference is modest, the authors should indicate how this might influence their observations.

7. In a similar study, albeit using EM, the Roux laboratory reported higher levels of anti-MPER Ab (4E10 and 2F5) binding to Env on the virion surface in the presence of bound sCD4 (Rathinakumar et al. JVI 2012). At a minimum, the authors need to discuss their results in the context of this previous study. It would also be worthwhile to determine whether sCD4 or a CD4 mimetic compound increases anti-MPER Ab binding in their approach.

Response to reviewers' comments:

Reviewer #1:

General remarks:

...For these reasons I fully support the publication of this study on Nature Communications.

We thank the referee for his/her firm support.

Minor points:

1- I was not able to find details about the image segmentation procedure implemented to measure the Area (in pixel) for Fig. 2 B. Please, provide it.

Details on this procedure are now provided as requested by the referee (see Methods *Image analysis* section, page 27 paragraph 1).

2- I suppose that the image are collected with a gated pulsed STED implementation. Since the manuscript highlight the methodology aspect of this studies, and in particular the STED microscopy method, I would like to read a little bit more details of the STED implementations.

Details on the specific STED implementations applied in this work are now provided in the “Methods” section (“STED microscopy measurements” subsection).

Suggestion:

3- Since the authors show a very multi-disciplinary aspect, from microscopy to image analysis, to biophysics, I will be happy to read in the conclusion a comment regarding what would be the next important steps from STED microscopy in the biological application addressed in this work.

Following referee's suggestion, we have added a comment at the end of the Discussion section. We suggest next possible steps in the application of STED to the problem of HIV infection, including its combination with advanced microscopy methods, such as molecular dynamic measurements by fluorescence correlation spectroscopy (STED-FCS) or membrane molecular order measurements using polarity sensitive dyes (STED-GP), or its combination with live-cell imaging.

Reviewer #2:

General Remarks:

...This study provides a nice example of how contemporary quantitative fluorescence microscopy can provide structural information under circumstances not amenable to conventional structural determination.

We thank the referee for his/her overall positive assessment on our work approach.

1. The authors have provided no explicit indication of whether the Fab binding sites are fully saturated during preparation of their samples. This is clearly important to ensure that their estimations of the number of Env clusters per virion are reliable. Showing that a higher concentration of Ab yields the same result would be helpful in this regard.

We agree with the referee that to ensure reliable estimations of the number of Env clusters per virion, it is important to show that a higher concentration of Ab yields the same result. Therefore, we had performed titration experiments up to antibody concentrations 5 times higher than those used in Figures 2c and d. As shown in new Supplementary Figure 1a, at higher Ab concentrations we observed foci distributions comparable to those displayed in Figures 2c and d. This is also explained now in the text (page 7, second paragraph).

2. The authors demonstrate that a panel of Abs that target either the MPER or diverse epitopes in gp120 all yield the same apparent distribution of Env... one might expect that 2G12 would indicate a distinct Env distribution, or perhaps a greater number of visible (Ab-bound) Envs per virion. This should be discussed.

We thank the reviewer for bringing this issue to our attention. The reviewer is right; 2G12 actually displays a greater number of Ab-bound per virion, even though Env foci distribution is similar. To address this question, in addition to new text and a new reference (see page 7, second paragraph), we have added the new Supplementary Figure 1b. Following referee's suggestion, we discuss that the observation of comparable foci distributions detected by anti-gp41 4E10/10E8 and anti-gp120 antibody 2G12 supports the idea that Env clustering is governed through the interactions established by Gag-interacting Env tail (again, see previous work by Chojnacki et al., refs: 25 and 26), and it appears to evolve independently of the Env oligomerization state.

3. How does colocalizing anti-MPER Abs with another Ab indicate "neutralization relevant binding events"? This is not clearly described.

Colocalization experiments described in the "Colocalization of antibodies..." subsection were meant to establish whether anti-MPER antibodies colocalized on the surface of virions with anti-gp120 antibodies that neutralize HIV-1 by binding to native Env. To avoid confusion, the first sentence of this subsection has been rephrased in the revised version of the manuscript.

4. Why are the correlation coefficients not higher? ...could this also reflect limited binding efficiencies? Or competitive binding of the two Abs? ...Interpretation of these results would be helped by measuring the pixel-wise correlation coefficient for Abs with

overlapping epitopes (ie, competitive Abs). This would indicate their full dynamic range in determination of the correlation coefficient.

We agree with the referee that the correlation coefficients are lower than expected. Following referee's recommendation we have obtained additional information on the dynamic range of our measurements, and established colocalization in a competitive binding experiment under our conditions. As expected from the competitive binding of the two Abs, the pixel-wise correlation coefficient decreased significantly with respect to self-colocalization controls ("Auto") and anti-gp120 samples. These data support that the values of the maximum correlation coefficients measured do not reflect competitive binding. These observations are now described in page 9 (second paragraph) and new Supplementary Figure 2, which also includes an explanatory diagram.

5. For the measurements of the number of Fabs per virion, the photon count distributions need to be calibrated using individual Fabs stuck to the surface in the absence of virus...

This control was already done, but not mentioned in the previous version of the manuscript. The outcome of this control (same number of photons per Fabs bound to virions or the support surface) is now included in the SI of the modified version (additional panel c of Supplementary Fig. 3).

6. The Trkola laboratory has reported that NL4-3 has somewhat more Env trimers per virion than JR-FL ... While the difference is modest, the authors should indicate how this might influence their observations.

Following referee's indication, we have added text and a new reference to account for those findings (i.e., estimated mean virion Env trimers of 13.5 and 11.8 for NL4-3 and JR-CSF, respectively), and discussed their potential implications in the "Accessibility to MPER..." subsection (page 12, first paragraph).

7. ... Roux laboratory reported higher levels of anti-MPER Ab (4E10 and 2F5) binding to Env on the virion surface in the presence of bound sCD4... At a minimum, the authors need to discuss their results in the context of this previous study.

Our main goal was to study MPER accessibility through STED in the context of intact virions, informing in that way the potential efficacy of native versions of Env as anti-MPER vaccines, and the structure-function features to be preserved in effective anti-MPER antibodies (see Introduction, last paragraph). We agree with the reviewer that investigating the effect of different ligands (fusion promoters and blockers) on the accessibility of neutralizing epitopes on Env, will be worthwhile to determine through STED, not only on intact virions, but also in the more physiologically relevant context of the virion-cell interaction (see also response to point 3 raised by the previous reviewer). In particular, studying the effect of the CD4 receptor and co-receptor binding in Env accessibility for bnAbs directly on cell membranes is part of one of our future projects. Following the referee's suggestion, we have rewritten the last paragraph of the "Discussion" section, to emphasize this possible course of the STED application to HIV research and to discuss our findings taking into account results by the Roux laboratory.

REVIEWERS' COMMENTS:

Reviewer #1 (Remarks to the Author):

I think the revised manuscript is significantly improved and does not need further review steps from my side. I strongly recommend publication.

I am just curious about one detail introduced in the description of the STED microscope. Authors states that the pulse of the STED beam is 3.5 ns, is it true? or is it a typo? In the first case, I will really appreciate if the author can cite some references regarding the time-gating STED approach, since in case of nanoseconds STED-beam pulse (i.e. quasi Continuous wave), time gating becomes fundamental to achieve the high-resolution values (60 nm) reported in the manuscript.

Reviewer #2 (Remarks to the Author):

The authors have done an excellent job responding to my concerns. I strongly support publication of their manuscript in Nature Communications.

Response to reviewers' comments:

Reviewer #1 (Remarks to the Author):

I think the revised manuscript is significantly improved and does not need further review steps from my side. I strongly recommend publication.

We thank the reviewer for the positive recommendation, and the thoroughness of his/her assessment.

I am just curious about one detail introduced in the description of the STED microscope. Authors states that the pulse of the STED beam is 3.5 ns, is it true? or is it a typo?

It was indeed a typo on our part when providing an updated Methods section on microscope setup. The correct value should read 800 ps which is a typical value used in pulsed-STED applications. This has been corrected in the revised version of the manuscript

Reviewer #2 (Remarks to the Author):

The authors have done an excellent job responding to my concerns. I strongly support publication of their manuscript in Nature Communications.

We thank the reviewer for his/her exhaustive assessment, which has been very helpful for producing a more solid version of the work.